# RobustMerge: Parameter-Efficient Model Merging for MLLMs with Direction Robustness

**Fanhu Zeng**[1]  **Haiyang Guo**[1]  **Fei Zhu**[2*]  **Li Shen**[3,4]  **Hao Tang**[5*]

[1]MAIS, Institute of Automation, Chinese Academy of Sciences
[2]Centre for Artificial Intelligence and Robotics, HKISI-CAS
[3]Shenzhen Campus of Sun Yat-sen University [4]Shenzhen Loop Area Institute
[5]State Key Laboratory of Multimedia Information Processing,
School of Computer Science, Peking University
`challengezengfh@gmail.com, guohaiyang2023@ia.ac.cn,`
`zhfei2018@gmail.com, shenli6@mail.sysu.edu.cn, haotang@pku.edu.cn`

## Abstract

Fine-tuning pre-trained models with custom data leads to numerous expert models on specific tasks. Merging models into one universal model to empower multi-task ability refraining from data leakage has gained popularity. With the expansion in data and model size, parameter-efficient tuning becomes the common practice for obtaining task-specific models efficiently. However, few methods are dedicated to efficient merging, and existing methods designed for full fine-tuning merging fail under efficient merging. To address the issue, we analyze from low-rank decomposition and reveal that ***direction robustness*** during merging is crucial for merging efficient modules. We furthermore uncover that compensating for the gap between stark singular values contributes to direction robustness. Therefore, we propose ***RobustMerge***, a training-free parameter-efficient merging method with complementary parameter adaptation to maintain direction robustness. Specifically, we **(1)** prune parameters and scale coefficients from inter-parameter relations for singular values to maintain direction stability away from task interference, and **(2)** perform cross-task normalization to enhance unseen task generalization. We establish a benchmark consisting of diverse multimodal tasks, on which we conduct experiments to certify the outstanding performance and generalizability of our method. Additional studies and extensive analyses further showcase the effectiveness. Code is available at `https://github.com/AuroraZengfh/RobustMerge`.

## 1 Introduction

Rapid development of foundation models has facilitated the construction of expert models from custom data. Modern models like large language models (LLMs) are pre-trained on various datasets to obtain general knowledge and employing pre-trained models typically involves fine-tuning on task-specific data to gain ability in specific areas. When dealing with tasks of different domains, multi-task learning [47] is a common paradigm to mitigate performance variations. However, particular knowledge may be required progressively over time. As the model becomes larger [4, 59], once the model is specialized on specific datasets, it is time-consuming and resource-intensive to retrain models to gain knowledge of another area, even encountering catastrophic forgetting [67]. Furthermore, issues regarding data privacy may obstruct its practical application. To address these issues, model merging [51] has been proposed to integrate multiple separate models of specific knowledge off-the-shelf into one model with multi-task ability without the demand of training or accessing data. Its effectiveness and convenience show great potential in various downstream tasks [10, 49].

---

[*]Corresponding authors.

39th Conference on Neural Information Processing Systems (NeurIPS 2025).

Despite its popularity, crucial problems for model merging remain unsolved, restricting its real-world deployment. First, with larger model sizes like multimodal large language models (MLLMs) and massive data, parameter-efficient tuning (PEFT) [16] has become the most popular and effective tuning approach for large models. However, existing model merging methods focus on full fine-tuning (FFT) techniques [62, 11], which struggle with distribution shift and undergo performance drops when directly applied to parameter-efficient model merging, as is illustrated in Fig. 1. Moreover, another issue lies in that current high-performance methods rely on extra information of seen tasks (*e.g.*, validation data [64], extra storage [18]) to boost the performance. Therefore, they can only handle seen tasks and fail to generalize to unseen tasks, questioning their robustness and extensibility in real-world scenarios, as is concluded in Tab. 1. The most related work is LoraHub [17]. However, its requirement for coefficient optimization through test-time adaptation severely hinders its application.

Inspired by the stated shortcomings, we aim to develop a merging algorithm for parameter-efficient modules with generalizability. First, we analyze the reason behind the performance drop. We observe (1) stark singular values and (2) a distinct wider distribution in efficient parameters that differ from full fine-tuning. Moreover, starting from the perspective of low-rank decomposition, we reveal that direction robustness, *i.e.*, maintaining directions of singular values, is crucial for efficient merging. From the above observations, we propose Robust-Merge, a novel parameter-efficient method for high-performance merging of multimodal large models and introduce effective complementary[2] parameter adaptation to maintain directions for performance enhancement. Concretely, we prune ineffective parameters and construct scaling coefficients from inter-parameter relations directly on LoRA components to mitigate interference between tasks aroused from stark singular values difference. Additionally, we perform cross-task normalization to balance tasks of different data scales and enhance unseen task gener-

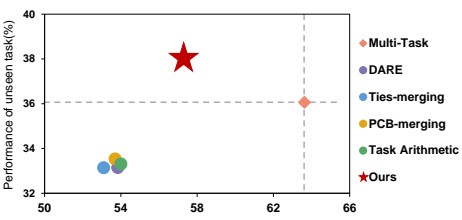

Figure 1: Performance balance between seen task enhancement and unseen task generalization.

Table 1: Prerequisites and application scope of different methods.

| Methods | Validation Free | Extra Storage Free | Unseen Tasks | Parameter-Efficient Merging |
|---|---|---|---|---|
| Task Arithmetic | ✗ | ✓ | ✓ | ✗ |
| DARE | ✓ | ✓ | ✓ | ✗ |
| Ties-merging | ✓ | ✓ | ✓ | ✗ |
| Pcb-Merging | ✓ | ✓ | ✓ | ✗ |
| LoraHub | ✗ | ✗ | ✓ | ✓ |
| AdaMerging | ✗ | ✓ | ✓ | ✗ |
| EMR-Merging | ✓ | ✗ | ✗ | ✗ |
| **RobustMerge** | ✓ | ✓ | ✓ | ✓ |

alization. It is notable that our method is free from any additional data or storage and does not require explicit decomposition, which equips the method with more flexibility and efficiency.

We conduct experiments on a benchmark consisting of eight seen tasks and four unseen tasks with diverse fields to evaluate the ability on multimodal generative tasks. We also report results on common evaluation benchmarks, and it shows that our method promotes both seen (**3.4%**), unseen tasks (**4.5%**) and comprehensive common ability with a substantial margin, demonstrating the effectiveness and generalizability of our method. We additionally perform experiments on vision tasks along with extensive analyses to validate the utility of our method. Our contributions are summarized as follows:

- We focus on parameter-efficient model merging, highlighting the necessity of high-performance parameter-efficient merging algorithms free from additional data or storage.
- We analyze from the perspective of direction robustness of singular values in low-rank decomposition and propose an effective training-free merging algorithm with complementary parameter adaptation to maintain direction for merging performance enhancement.
- We conduct extensive experiments and achieve superior results compared to existing approaches, which strongly validates the effectiveness and generalizability of the method.

## 2  Related Work

**Multimodal large language models.** With the surge in data volume and model size, large language models (LLMs) [46, 56, 1] have shown their powerful performance. They are constructed with decoder-only blocks and respond to inputs in an auto-regressive way, which shows their potential

---

[2]In contrast to **individual**, we use the term to distinguish between subspace multiplication (along $r$ dimension) and original multiplication (along $d_i/d_o$ dimension) of matrix.

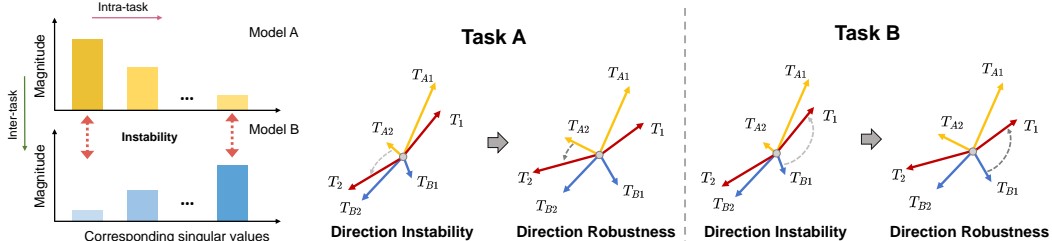

Figure 2: Illustration of merging A and B in low-rank space for evaluation of each task. The magnitude of vector represents the numerical singular value. Left: Stark singular values exist within task, leading to instability when merging between tasks. Right: As directions of large singular value are naturally robust, direction instability is more likely to happen for small values when merging specific singular vectors. Scaling tail values contributes to direction robustness and promotes the performance.

in both classification [57] and generative tasks [7]. Furthermore, multimodal large language models (MLLMs) enhance large models with vision perception ability. They obtain visual features with a vision encoder and align image-text features with a cross-modality module [32, 26] and so on. Current research on large models is dedicated to directly fine-tuning one independent model with task-specific data to get better results. Rather than improving the performance of a certain domain, we focus on integrating models into one model to boost efficiency and handle multiple tasks simultaneously.

**Parameter-efficient tuning.** When fine-tuning a pre-trained model with task-specific data, training the whole model would not only disrupt the representations obtained from billions of data but also become resource-intensive. To address the issue, parameter-efficient tuning [13] is introduced to refrain from fine-tuning the whole model. It typically trains lightweight modules to make the model adapt to downstream tasks and achieves competitive results compared to full fine-tuning models. Various efficient tuning techniques have been explored, like prompt learning [20, 23, 66], adapter learning including LoRA [16, 60, 34, 41, 33], (IA)$^3$ [31] and so on. In this paper, we focus on LoRA, as it is the most commonly utilized PEFT method and has demonstrated its usefulness in various fields [68, 9] especially for large models [32].

**Model merging.** Model merging [63, 52, 27] refers to merging multiple models of different capabilities to handle multi-task learning with one universal model [21, 40]. Task Arithmetic [19] presents a paradigm that obtains task vectors from subtracting a pre-trained model from fine-tuned models and treats model merging as arithmetic operations of task vectors. It has gained widespread attention in various fields [53]. Ties-merging [62] trims and elects signs to reduce interference. DARE [65] randomly drops parameters and rescales the remaining ones to approximate the original embedding. PCB-merging [11] introduces parameter adjustment with competition balancing to address potential conflicts. However, most of them focus on merging models with FFT on classification tasks [5], and the distribution shift prevents their ability to acquire satisfying performance [54]. Some recent works [29, 30] also focus on merging checkpoints during pre-training to enhance downstream performance. By contrast, we concentrate on parameter-efficient merging with multimodal tasks.

## 3 Methodology

We first describe basic notations for efficient merging, then show our observation and motivation for reducing task interference when merging efficient modules, and finally introduce our method to improve the performance of parameter-efficient merging for multimodal large language models.

### 3.1 Preliminary and Notations

**Parameter-efficient tuning** keeps the pre-trained model frozen and fine-tunes a lightweight module to adapt to downstream tasks. In this paper, we focus on LoRA [16], a low-rank adaptation technique that decomposes additional parameters into two low-rank matrices. Formally, for a weight matrix $\mathbf{W_0} \in \mathbb{R}^{d_o \times d_i}$, the updated matrix is depicted as:

$$\mathbf{W} = \mathbf{W_0} + \Delta\mathbf{W} = \mathbf{W_0} + \mathbf{B} \cdot \mathbf{A}, \tag{1}$$

where $\mathbf{B} \in \mathbb{R}^{d_o \times r}$, $\mathbf{A} \in \mathbb{R}^{r \times d_i}$ and rank $r \ll \min(d_i, d_o)$.

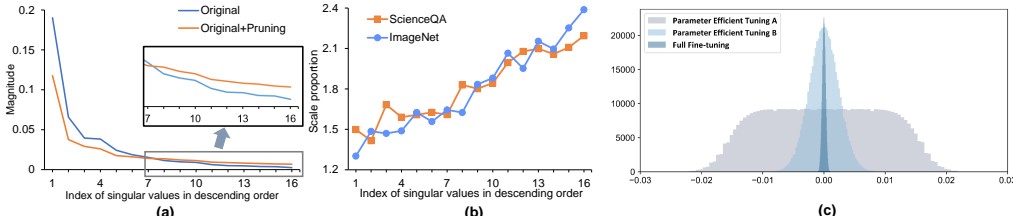

Figure 3: (a) Magnitude of singular values for original and pruned matrix. Stark singular values are observed in original matrix and pruning effectively scale tail ones. (b) Effectiveness of RobustMerge by adaptively reducing interference with larger scale on smaller singular values. (c) Distribution of FFT and PEFT modules. Parameters of FFT, and different components in efficient tuning have different distributions.

**Model merging** targets at combining multiple models of the same structure $\{\theta_1, \cdots, \theta_N\}$ that are fine-tuned from pre-trained model $\theta_{pre}$ into one new model $\theta_m$ and maintaining multi-task ability in a training-free manner. Existing full fine-tuning (FFT) methods follow the paradigm proposed by Task Arithmetic (TA) [19]. Typically, they construct the task vector by performing a subtraction operation $\tau_n = \theta_n - \theta_{pre} \in \mathbb{R}^d$, conduct a merging algorithm on the task vector subspace, and obtain the final merged model by adding the pre-trained model, *i.e.*, $\theta_m = \theta_{pre} + \lambda \sum_{n=1}^{N} \Phi(\tau_n)$, where $\Phi(\cdot)$ stands for the merging algorithm.

**Parameter-efficient model merging** differs from the traditional model merging, as the backbone of the foundation model is frozen and the updated matrices to be merged are randomly initialized. Consequently, we use $\Delta\mathbf{W}$ to represent merging modules for simplification, and exploit the model merging method on these parameter-efficient modules, *i.e.*, $\mathbf{W}_m = \mathbf{W_0} + \lambda \sum_{n=1}^{N} \Phi(\Delta\mathbf{W}_n)$.

### 3.2 Motivation and Observation

While existing methods perform well on FFT merging, challenges remain unsolved when it comes to PEFT merging with suboptimal performance. To have a better understanding of the difference, we (1) first analyze the parameter distribution and low-rank decomposition of a single task, (2) then reveal key factors for parameter-efficient merging of multiple tasks, and (3) finally propose an effective merging algorithm for parameter-efficient modules built on these observations.

**Direction robustness is crucial for merging models of multiple tasks.** To illustrate the uniqueness of parameter-efficient tuning in merging compared to full fine-tuning, *i.e.*, the low-rank matrices, we decompose them using singular value decomposition (SVD) to obtain singular values with corresponding directions and introduce the notation of ***Direction Robustness*** that plays a vital role in merging. Concretely, for a single matrix, the direction for each singular value can be viewed as task-specific knowledge in low-rank space and the magnitude of the singular value is the extent to which the knowledge is utilized in the current task. Theoretical analysis is provided in Appendix A.

We visualize the distribution of singular values for efficient modules in Fig. 3a and observe a stark difference between head and tail singular values (intra-task). Therefore, for models of diverse tasks (inter-task) to be merged, directions of large singular values are inherently prone to direction change. When merging models and evaluating on a certain task, *i.e.*, task-specific knowledge, the corresponding small singular values are more likely to alter the direction, challenging the stability. The same direction instability appears on other singular vectors when the evaluated task changes. Therefore, direction robustness, which refers to maintaining the direction of each singular vector during low-rank matrix merging, is crucial for reducing task interference, as each of them represents task-specific knowledge and contributes to merging performance. We give an illustration of merging models fine-tuned on specific tasks A, B and evaluating on each task separately in Fig. 2.

**Mitigating gap between singular values is effective for high-performance merged model.** As different tasks have their principal singular directions, certain directions may possess large singular values in one task and small ones in another. Based on the observation above and in Fig. 3, it can therefore be inferred that the direction of tail singular values for certain tasks is more likely to cause instability when merging, and mitigating the gap is crucial for resolving the interference between different tasks. One direct way is to adaptively scale tail values, which has less impact on vectors with larger singular values, while greatly contributing to small ones. This can be confirmed by Fig. 3a

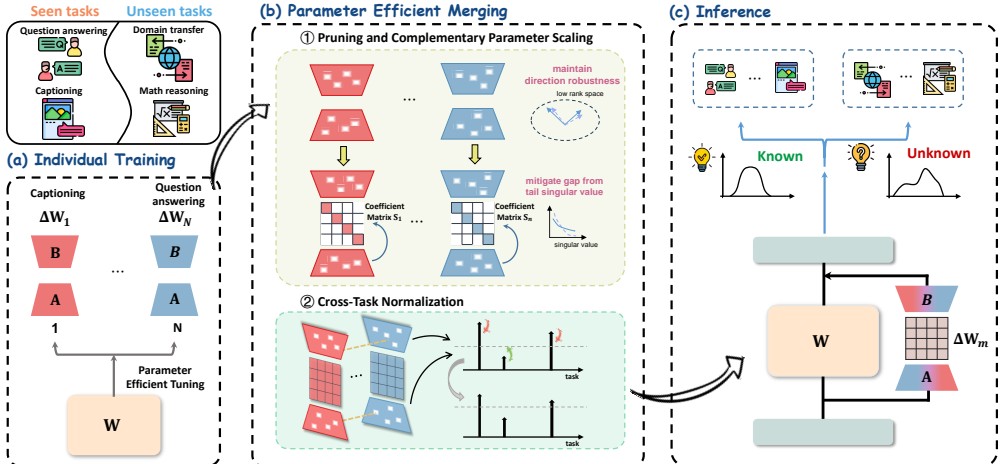

Figure 4: Diagram of RobustMerge. Tasks are divided into seen and unseen ones. Checkpoints of seen tasks are trained employing the standard individual training and are merged following the pipeline of inter-parameter adaptation. During inference, the merged model is required to both enhance seen tasks and be generalizable to unseen tasks with an unknown distribution.

and Fig. 3b, which clearly show that our method introduced in Sec. 3.3 changes the distribution of singular values and adaptively adjusts the singular values by scaling smaller singular values by a larger multiple, thereby alleviating direction instability and achieving better performance. Detailed illustration of singular values is shown in Appendix B.

**Parameters of efficient modules have distinct distributions.** We also depict the distribution of elements in Fig. 3c to figure out the difference between two types of merging. It can be found that most parameters of full fine-tuning have a much smaller and concentrated distribution (distribution in dark blue), where the problem of sign conflict becomes particularly prominent [62]. Conversely, parameters in efficient components have a relatively wider range of distribution (light blue and gray), and direction instability rather than sign conflict is the main issue for interference between tasks, which we give a detailed comparison in Sec. 4.3.

**Parameter-efficient modules have intrinsic relations.** The two LoRA matrices have asymmetric functions in PEFT [69, 55]. As pointed out by AsymmetryLoRA [69], a random untrained $\mathbf{A}$ performs as well as a fine-tuned one and $\mathbf{B}$ improves the bound. HydraLoRA [55] reveals that shared $\mathbf{A}$ can reserve knowledge. To determine the distinct function of the two matrices in merging, we depict the distribution of $\mathbf{A}$, $\mathbf{B}$ respectively in Fig. 3c. It turns out that $\mathbf{B}$ follows a Gaussian distribution and $\mathbf{A}$ has an approximately uniform distribution. It corresponds with existing research that $\mathbf{B}$ also has a more unique and crucial role in PEFT merging. Due to their expression, we aim to scale the singular value directly on two LoRA modules to avoid explicit and time-consuming decomposition.

### 3.3 RobustMerge: Parameter-Efficient Model Merging for MLLMs

Motivated by the observations, we introduce a novel model merging method for parameter-efficient components, which targets maintaining direction robustness and compensating for the gap between singular values by adaptively scaling tail singular values. As is illustrated in Fig. 4, our approach is divided into **pruning and complementary parameter scaling** and **cross-task normalization**.

**Pruning and complementary parameter scaling.** Due to significantly wider distributions, changing larger parameters are more likely to alter directions in low-rank space. Therefore, rather than electing parameters of the same sign [62, 18] with delicate design, we simplify the definition of ineffective parameters to be those with small values in magnitude. In this way, the direction of matrices is not greatly changed by reserving larger parameters, and knowledge of the specific task is therefore retained when mitigating interference between models of different tasks. Consequently, with $\mathcal{M}(\cdot)$ as the binary operation matrix, the updated matrices can be formulated as:

$$\widetilde{\mathbf{A}} = \mathcal{M}_{\mathrm{A}}(k) \odot \mathbf{A}, \quad \widetilde{\mathbf{B}} = \mathcal{M}_{\mathrm{B}}(k) \odot \mathbf{B}, \tag{2}$$

where $\odot$ stands for element-wise multiplication, and $k$ is the pruning rate of parameters. Formally, it sets $k$ percentage of parameters with small values sorted by magnitude to zero. Fig. 3a shows its utility in altering distribution and scaling tail singular values.

After pruning ineffective parameters, the remaining ones should be refined to scale tail singular values and complement the performance gap caused by task interference. Considering that explicitly operating on decomposed matrices is time-consuming, we directly adjust original low-rank matrices to achieve the same goal inspired by the asymmetry and correlation between two LoRA modules stated in Sec. 3.2 and that $\mathbf{A}$ is almost orthogonal in high-dimensional space. Specifically, we propose to adaptively adjust singular values through complementary parameter scaling for transforming $\mathbf{B}$ to compensate for performance deficiencies resulting from the gap between singular values. As $\mathbf{A}$ follows a uniform distribution, we construct scaling coefficients from the statistical characteristics of $\mathbf{A}$ for singular values. We define the scaling matrix $\mathbf{S}$ as a diagonal matrix, and the elements on the diagonal are:

$$\mathbf{S}^i = \frac{\sum_{j=1}^{d_i} \mathrm{abs}(\mathbf{A}_{[i,j]})}{\sum_{j=1}^{d_i} \mathrm{abs}(\mathcal{M}_{\mathrm{A}[i,j]} \odot \mathbf{A}_{[i,j]})}, \quad i = 1, \cdots, r. \tag{3}$$

This can be viewed as singular value adaptation in low-rank space, in which small values of each model are effectively increased in larger proportions (Fig. 3b), thereby contributing to minimizing task conflicts aroused by direction instability while refraining from explicit computation of decomposition.

**Cross-task normalization.** Complementary parameter scaling coefficient $\mathbf{S}$ is determined in a task-independent manner. On the one hand, the imbalance in data size between different seen tasks leads to overfitting for data-abundant tasks and underfitting for data-scarce tasks. Additionally, it also poses a negative effect on the generalization to unseen tasks. Consequently, we conduct normalization on scaling matrices across all tasks to reduce the impact of coefficient imbalance, mathematically formulated as:

$$\widetilde{\mathbf{S}}_n^i = \mathbf{S}_n^i / \sum_{n=1}^{N} \mathbf{S}_n^i, n = 1, \cdots, N. \tag{4}$$

The normalization provides more balance across diverse types of tasks and therefore achieves more stable performance. It also enhances the ability on unseen tasks, which is shown in Fig. 5c. The final efficient parameter can be rewritten as follows:

$$\Delta \widetilde{\mathbf{W}} = \lambda \sum_{n=1}^{N} (\widetilde{\mathbf{B}}_n \cdot \widetilde{\mathbf{S}}_n) \cdot \sum_{n=1}^{N} \widetilde{\mathbf{A}}_n, \tag{5}$$

and the merged model weights can be obtained by adding merged parameter-efficient modules of all tasks. It should be emphasized

---

**Algorithm 1** Procedure of parameter-efficient merging with complementary parameter adaptation.

**Input:** Fine-tuned models $\{\mathbf{A}_n, \mathbf{B}_n\}_{n=1}^{N}$, pruning rate $k$, rank $r$ and $\lambda$
**Output:** Merged Parameter-Efficient Model $\mathbf{W}$
▷ Step 1: Pruning and Complementary Parameter Scaling.
  $\mathcal{M}_{\mathrm{A}}(k) = \mathrm{binary}(\mathrm{set\_topk\_nonzero}(\mathbf{A}, k))$
  $\mathcal{M}_{\mathrm{B}}(k) = \mathrm{binary}(\mathrm{set\_topk\_nonzero}(\mathbf{B}, k))$
  $\widetilde{\mathbf{A}} = \mathcal{M}_{\mathrm{A}}(k) \odot \mathbf{A}$
  $\widetilde{\mathbf{B}} = \mathcal{M}_{\mathrm{B}}(k) \odot \mathbf{B}$
**forall** $i = 1, \cdots, r$ **do**
  $\mathbf{S}^i = \sum_{j=1}^{d_i} \mathrm{abs}(\mathbf{A}_{[i,j]}) / \sum_{j=1}^{d_i} \mathrm{abs}(\mathcal{M}_{\mathrm{A}[i,j]} \odot \mathbf{A}_{[i,j]})$
**end**
▷ Step 2: Cross-Task Normalization.
**forall** $n = 1, \cdots, N$ **do**
  $\widetilde{\mathbf{S}}_n^i = \mathbf{S}_n^i / \sum_{n=1}^{N} \mathbf{S}_n^i$
**end**
▷ Obtain parameter-efficient modules.
**forall** $n = 1, \cdots, N$ **do**
  $\widetilde{\mathbf{S}}_n = \mathrm{Diag}(\widetilde{\mathbf{S}}_n^i)$
**end**
▷ Merge parameter-efficient modules.
  $\Delta \widetilde{\mathbf{W}} \leftarrow \lambda \sum_{n=1}^{N} (\widetilde{\mathbf{B}}_n \cdot \widetilde{\mathbf{S}}_n) \cdot \sum_{n=1}^{N} \widetilde{\mathbf{A}}_n$
  $\mathbf{W} \leftarrow \mathbf{W_0} + \Delta \widetilde{\mathbf{W}}$
**return** $\mathbf{W}$

---

that during the whole merging process, no validation data or extra information storage of seen tasks is required, certifying the superiority of the method.

## 4 Experiments

**Implementation details.** We conduct experiments on multimodal generative tasks, unseen task generalization, and vision tasks using multimodal models [32, 45]. We comprehensively extend our approach on the scale of the model, number of tasks, rank, and so on to certify the utility. Unless otherwise stated, all models are trained with a rank of 16. More details can be found in Appendix C.

**Datasets and baselines.** For multimodal task merging, we establish a MultiModal Merging Benchmark (MM-MergeBench), which comprises eight multimodal generative tasks including ScienceQA [36], ImageNet [5], VQAv2 [7], REC-COCO [22, 39], OCRVQA [42], Flickr30k [44],

Table 2: Performance of MM-Merge-Bench on eight seen and four unseen tasks.

| Method | SciQA | Image | VQA | REC | OCR | VizWiz | Flickr | IconQA | Average | AVQA | Image-R | S2W | TabMWP | Average |
|---|---|---|---|---|---|---|---|---|---|---|---|---|---|---|
| | **Seen Tasks** | | | | | | | | | **Unseen Tasks** | | | | |
| Individual | 83.74 | 96.02 | 67.58 | 43.40 | 65.50 | 64.80 | 57.29 | 75.54 | 69.23 | - | - | - | - | - |
| Zero-Shot | 61.73 | 40.87 | 62.88 | 36.10 | 41.16 | 41.03 | 49.07 | 14.09 | 43.37 | 51.62 | 28.27 | 5.98 | 15.01 | 25.22 |
| Multi-Task | 76.90 | 74.08 | 67.05 | 35.98 | 65.37 | 66.67 | 56.09 | 66.87 | 63.62 | 76.33 | 41.39 | 8.34 | 18.20 | 36.06 |
| Task Arithmetic | 71.94 | 57.49 | 67.06 | 38.90 | 62.87 | 44.80 | 49.20 | 39.21 | 53.93 | 74.78 | 37.37 | 7.52 | 13.57 | 33.31 |
| DARE | 71.59 | 57.25 | 66.26 | 39.38 | 62.56 | 44.93 | 49.13 | 39.59 | 53.84 | 73.75 | 37.67 | 7.56 | 13.62 | 33.15 |
| Ties-merging | 71.49 | 55.88 | 66.73 | 39.67 | **65.12** | 44.35 | 47.06 | 34.46 | 53.09 | 73.43 | 38.44 | 7.47 | 13.23 | 33.14 |
| PCB-merging | 71.10 | 57.82 | **67.59** | 38.22 | 64.35 | 44.58 | 48.90 | 37.01 | 53.70 | 74.57 | 36.28 | 7.84 | 15.44 | 33.53 |
| **RobustMerge** | **73.43** | **65.54** | 67.20 | **44.80** | 62.97 | **46.61** | **52.80** | **45.90** | **57.33** | **79.30** | **45.79** | **9.23** | **17.62** | **37.99** |

VizWiz-caption [12], IconQA [38]. It includes diverse multimodal tasks across various areas like question answering, grounding, classification, captioning and can comprehensively evaluate the performance of different merging methods in generative tasks. To demonstrate the generalizability on unseen tasks, we evaluate the merged models on four diverse datasets, ImageNet-R [15], AOKVQA [48], Screen2Word [58], TabMWP [37]. Detailed interpretation can be found in Appendix E. Besides, we also evaluate on general benchmarks like POPE [28], MME [6] and MMBench [35]. Experiments on vision tasks are provided in Sec. 4.2 and more results are shown in Appendix F.

For comparison methods, we re-implement Task Arithmetic [19], Ties-merging [62], DARE [65] and PCB-merging [11] on parameter-efficient modules of MLLMs to have a fair comparison. Detailed information about these baselines can be found in Appendix D.

## 4.1 Experiments on MLLM with Generative Tasks

We systematically evaluate parameter-efficient merging methods on multimodal generative tasks, in which LLaVA [32] is used as the foundation model, with CLIP-L-336 [45] as the image encoder.

**RobustMerge is effective in parameter-efficient tuning.** We evaluate the performance of various model merging methods. Concretely, we obtain independent models from fine-tuning separate datasets and merge models without re-accessing data. It is indicated from the left part of Tab. 2 that existing approaches suffer from a severe performance drop when merging parameter-efficient models, even worse than zero-shot in some cases. Also, they do not necessarily perform better than simple Task Arithmetic, showcasing the challenge in PEFT merging. By contrast, our method achieves superior results, consistently and substantially outperforming all previous methods by a solid margin (**3.4%** improvements on average). Notably, our approach even achieves comparable performance with multi-task learning. These results strongly validate the effectiveness of the method.

**RobustMerge enhances performance on unseen tasks.** Generalizability is crucial for evaluating merging methods as domain shifts are unavoidable and frequently occur in real-world scenarios. On the right of Tab. 2, we report merging performance directly evaluated on four unseen tasks. It is harder as the merged models have no clue for the distribution of unseen tasks. This is further confirmed by the poor performance of existing merging methods (TA, DARE, Ties), which is even worse than zero-shot on some occasions. Conversely,

Table 3: Performance of different merging models on general multimodal benchmarks.

| Method | POPE | MME | MMBench |
|---|---|---|---|
| Zero-Shot | 86.4 | 1476.9 | 66.1 |
| Traditional MTL | 86.9 | 1433.5 | 62.9 |
| Task Arithmetic | 87.0 | 1465.2 | 67.3 |
| DARE | 86.4 | 1475.7 | 67.4 |
| Ties-merging | 86,7 | 1489.4 | 66.6 |
| PCB-merging | 86.6 | 1490.7 | 66.3 |
| **RobustMerge** | **87.2** | **1494.9** | **68.1** |

our method significantly enhances performance with substantial **4.5%** average improvements and even outperforms multi-task learning. Notably, our method successfully promotes domain transfer (ImageNet-R) and specific knowledge task (TabMWP), further demonstrating the generalizability.

**RobustMerge outperforms on general multimodal benchmarks.** We additionally report results on general multimodal benchmarks POPE [28], MME [6] and MMBench [35] in Tab. 3 to evaluate

Table 4: Results of merging eight vision tasks with CLIP-ViT-B-32 as pre-trained foundation model.

| Method | Cars | MNIST | EuroSAT | GTSRB | DTD | RESISC45 | SUN397 | SVHN | Average |
|---|---|---|---|---|---|---|---|---|---|
| Zero-Shot | 59.7 | 48.5 | 62.3 | 32.6 | 60.7 | 43.8 | 45.5 | 31.4 | 48.0 |
| Individual | 74.3 | 99.3 | 65.2 | 92.9 | 88.7 | 58.4 | 99.1 | 96.4 | 84.2 |
| Task Arithmetic | 60.3 | 52.3 | 63.2 | 37.6 | 62.8 | 44.0 | 50.9 | 37.6 | 51.1 |
| DARE | 60.4 | 52.4 | 63.1 | 37.5 | 62.8 | 44.0 | 50.3 | 37.7 | 51.0 |
| Ties-merging | 60.7 | 56.4 | 62.4 | 33.9 | 61.3 | 43.1 | 51.1 | 42.9 | 51.5 |
| **RobustMerge** | **61.4** | **65.0** | **65.0** | **43.1** | **63.3** | **44.7** | **52.2** | **52.4** | **55.9 (+4.4)** |

Table 5: Results of merging eight vision tasks when pre-trained model scales to CLIP-ViT-L-14.

| Method | Cars | MNIST | EuroSAT | GTSRB | DTD | RESISC45 | SUN397 | SVHN | Average |
|---|---|---|---|---|---|---|---|---|---|
| Zero-Shot | 77.7 | 76.3 | 66.8 | 50.5 | 71.0 | 55.3 | 59.9 | 58.4 | 64.4 |
| Individual | 99.7 | 99.4 | 80.0 | 97.2 | 95.8 | 70.3 | 98.6 | 97.9 | 92.4 |
| Task Arithmetic | 78.6 | 79.7 | 68.5 | 53.6 | 73.5 | 55.8 | 65.7 | 60.9 | 67.0 |
| DARE | 79.5 | 81.4 | 68.8 | 56.5 | 75.0 | 56.6 | **65.8** | 62.8 | 68.3 |
| Ties-merging | 79.4 | **83.4** | 69.5 | 59.4 | 76.0 | 55.7 | 64.0 | 64.4 | 68.9 |
| **RobustMerge** | **79.7** | 82.8 | **70.6** | **62.4** | **78.4** | **58.2** | 64.7 | **70.3** | **70.9 (+2.0)** |

base capabilities of merged models like hallucination and so on. It shows that multi-task learning achieves inferior results, indicating the challenge. By contrast, our method enhances zero-shot performance, substantially outperforms existing methods, and retains common knowledge on challenging benchmarks with outstanding outcomes, certifying the effectiveness.

## 4.2 Experiments on VLM with Vision Tasks

For vision language model (VLM) with vision tasks, we follow the experimental setup outlined by Task Arithmetic [19] and fine-tune eight models with LoRA on corresponding vision datasets. The datasets consist of Cars [24], MNIST [25], EuroSAT [14], GTSRB [50], DTD [3], RESISC45 [2], SUN397 [61] and SVHN [43]. See Appendix E for details.

**RobustMerge is effective in vision tasks.** We evaluate our methods on CLIP-ViT-B-32 [45] and showcase the results in Tab. 4. It is indicated that when fine-tuning models with parameter-efficient techniques, previous methods do not observe significant improvements against zero-shot performance, questioning their utility in vision task-efficient tuning. By contrast, our method obtains a considerable 7.9% promotion against the ability of zero-shot and outperforms previous merging methods by a substantial margin (4.4%). It strongly validates the effectiveness of our method when merging vision models in a parameter-efficient way.

**RobustMerge scales well to large VLM models.** We also apply our method to larger models to certify the scalability of the method. Concretely, we fine-tune CLIP-ViT-L-14 on eight vision tasks separately and evaluate models with merged parameter-efficient components. The results in Tab. 5 exhibit that the performances of all methods improve with larger foundation models. Furthermore, RobustMerge achieves the best results with a 2.0% average improvement, demonstrating superiority.

Table 6: Influence of each component. Prune&scale and norm refer to pruning and complementary scaling.

| Prune&Scale | Norm | SciQA | Image | VQA | REC | OCR | VizWiz | Flickr | IconQA | Average |
|---|---|---|---|---|---|---|---|---|---|---|
| | | 71.94 | 57.49 | 67.06 | 38.90 | 62.87 | 44.80 | 49.20 | 39.21 | 53.93 |
| ✓ | | 73.03 | 64.18 | **67.50** | 43.12 | 58.19 | 46.36 | 52.24 | 44.54 | 56.14 (+2.21) |
| ✓ | ✓ | **73.43** | **65.54** | 67.20 | **44.80** | **62.97** | **46.61** | **52.80** | **45.90** | **57.33 (+3.40)** |

## 4.3 Ablation Study and Further Analysis

**Effectiveness of each component.** We progressively apply key components of our method, *i.e.*, pruning and complementary parameter scaling and cross-task normalization, to substantiate their effectiveness. Results in Tab. 6 illustrate that pruning and complementary parameter scaling fundamentally contribute to direction robustness and mitigating interference in model merging, and integrating them all further achieves more advanced results.

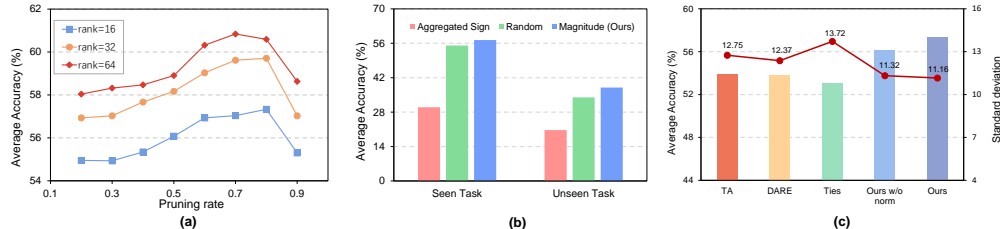

Figure 5: (a) Average performance of merging models with different pruning rates. Non-zero parameters decline according to the ineffective parameter criteria as the pruning rate increases. (b) Performance of different pruning techniques averaged on seen and unseen tasks. The aggregated sign achieves poor performance. (c) Comparison of average performance and standard deviation with existing methods. Cross-task normalization enhances performance with stable deviation.

**Impact of rank.** To explore the performance as the rank of LoRA varies, we train parameter-efficient components on different ranks. Results in Fig. 6 illustrate that the model obtains promotion from the improvements of rank, which increases the storage of knowledge in the update subspace. Moreover, our approach continuously outperforms existing methods by a substantial margin (3.4% in 16 and 3.3% in 128), validating the scalability of the method.

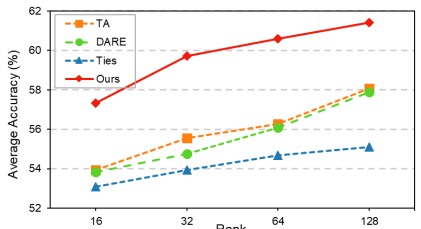

Figure 6: Average performance with different ranks.

**Influence of pruning rate.** As is revealed in Sec. 3.2, parameters of small values have less influence on the direction robustness of low-rank decomposition. Therefore, we prune parameters according to the magnitude to facilitate the merging procedure. To further validate this point of view, we gradually increase the pruning rate and show the variation of average performance in Fig. 5a. It is elucidated that: (1) As the pruning rate increases, the performance gradually boosts due to reduced interference between tasks; (2) Finally, the performance undergoes a sharp decrease as pruning larger parameters significantly influences directions and task knowledge. Consequently, the results are in accordance with the aforementioned analysis, underlining the utility of the ineffective parameter pruning strategy in our method.

**Parameter pruning in magnitude provides superior results.** We additionally compare with pruning techniques employed in DARE [65] and Ties-merging [62] to showcase the effectiveness of the proposed parameter pruning technique. Concretely, we employ random pruning and aggregated sign as pruning criteria, respectively, and evaluate their performance. The results in Fig. 5b reveal that sign conflict is not crucial in efficient merging, with performance worse than random. By contrast, our magnitude-based parameter pruning technique achieves better results in multimodal tasks and outperforms existing

Table 7: Influence of complementary parameter scaling. Coefficients solely dependent on specific module ($\mathbf{A}$, $\mathbf{B}$ or both) perform inferior to adaptive coefficients with inter-parameter relations.

| Method | Seen Tasks | Unseen Tasks |
|---|---|---|
| Baseline (w/o adaptation) | 54.1 | 32.5 |
| Ours (individual, A) | 54.5 (+0.4) | 32.1 (−0.4) |
| Ours (individual, B) | 55.4 (+1.3) | 34.1 (+1.6) |
| Ours (individual, A + B) | 51.7 (−2.4) | 35.0 (+2.5) |
| **Ours (inter-parameter)** | **57.3 (+3.2)** | **38.0 (+5.5)** |

approaches by a substantial margin (2.3% and 4.0%, respectively). It indicates that the value in magnitude, other than sign interference, plays a vital role in parameter-efficient model merging. We attribute the promotion to a significantly wider distribution of parameter-efficient models than full fine-tuning models, and pruning according to sign inevitably changes direction in low-rank space. Conversely, our method avoids task conflicts with less impact on principal direction.

**Complementary parameter scaling effectively compensates for performance drop.** It is elucidated in Sec. 3.3 that we construct coefficients with influence interwoven between parameters. To figure out its effectiveness, we replace it with different scaling strategies. Concretely, we decouple the interaction between the two modules, employing coefficients from $\mathbf{A}$, $\mathbf{B}$, individually. We additionally conduct scaling for $\mathbf{A}$ and $\mathbf{B}$ simultaneously and report quantitative results in Tab. 7. The study certifies the benefit of scaling coefficients from complementary parameter adaptation, and adaptively adjusting coefficients of $\mathbf{B}$ effectively promotes performance, which is in accordance with the analysis above. It is also demonstrated that the performance does not necessarily become better by utilizing more complex scaling coefficients (2.4% decrease in seen tasks).

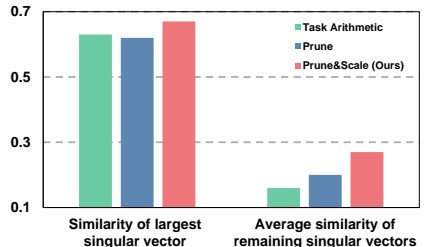 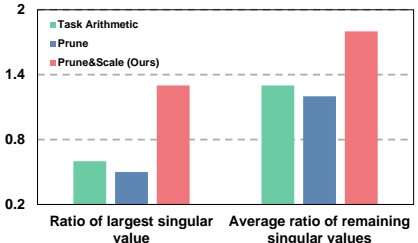

Figure 7: Similarity of singular vector and ratio of singular value on different merging techniques.

**Cross-task normalization provides more stable performance.** As is illustrated in Sec. 3.3, cross-task normalization provides not only consistent and stable performance on seen tasks but also advanced promotion on unseen tasks. We analyze the correlation between performance and variance in Fig. 5c. Concretely, compared with existing methods, our approach achieves 3.4% better performance (57.3% v.s. 53.9%) with significantly smaller variance (**1.2%**). Notably, employing cross-task normalization strengthens the advantage. Specifically, it improves 1.2% average performance while reducing 0.2% on standard deviation, showcasing the superiority of the proposed method.

**Further analysis of direction robustness.** We formulate the evaluation metrics for measuring direction robustness to better understand parameter-efficient model merging. We use the *average similarity of each corresponding singular vector* between task-specific models and merged models as the criterion, which reflects the direction deviation in merging. Larger similarity means the direction possesses more robustness and is not prone to changing direction during merging, thereby maintaining the performance of a specific task. Moreover, we also incorporate the *ratio of singular value* between merged and original models to comprehensively reflect the degree of specific knowledge learning. As is depicted in Fig. 3, the largest singular value displays more direction robustness, so we divide the value into the largest and the average of the remaining parts in Fig. 7 for better illustration of different model merging approaches.

It can be concluded that: (1) During merging, the largest vector tends to be stable, while remaining vectors are extremely dissimilar (direction instability), which leads to a performance drop in evaluation. By contrast, our method substantially improves the similarity of remaining vectors, strongly promoting merging performance; (2) The results of the ratio of value also reflect that for a specific model, existing methods would decrease the largest singular value and fail to sufficiently strengthen smaller singular values. By contrast, our method better enhances smaller values and maintains task-specific knowledge during merging, which is consistent with our view that scaling smaller values contributes to direction robustness. Furthermore, these analyses also give a clear explanation about the function of each component in the proposed method: (1) Prune is to resolve the interference between tasks while exhibiting the least influence on the direction, and the sparsification also boosts the robustness of small values; (2) Scaling after prune aims to compensate for the singular value drop raised by pruning, thereby enhancing the direction robustness.

## 5  Conclusion

This paper focuses on parameter-efficient model merging for large foundation models. We analyze from low-rank decomposition and reveal that direction robustness is crucial for merging efficient modules. We furthermore uncover that scaling tail singular values can effectively mitigate task interference and maintain direction robustness. Therefore, we introduce RobustMerge, an effective merging technique to maintain directions in low-rank space. We conduct extensive experiments and comprehensive analyses to showcase the superiority and scalability of the approach. This is the first attempt at parameter-efficient model merging from the perspective of direction robustness, and we hope it can inspire more advanced parameter-efficient merging methods.

**Limitations and future work.** We do not validate the method on more structures and tasks. However, since our method is a model-agnostic and task-agnostic post-processing algorithm, this will not be a bottleneck given numerous models on platforms like Huggingface. Also, we propose the concept of direction robustness in parameter-efficient merging, but we do not design a specific algorithm on decomposed matrices for the purposes of efficiency. We believe they would be promising directions that are left for future development in various downstream areas of parameter-efficient learning.

## Acknowledgments and Disclosure of Funding

This work is partially supported by the Fundamental Research Funds for the Central Universities, Peking University, and the InnoHK program. Li Shen is supported by the NSFC Grant (No. 62576364), Shenzhen Basic Research Project (Natural Science Foundation) Basic Research Key Project (No. JCYJ20241202124430041).

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

# Appendix

## A   Theoretical Analysis of Singular Value Decomposition in Merging

As we focus on the merging of low-rank matrices, we first introduce basic notations of singular value decomposition and then describe its application in merging.

### A.1   Background of Singular Value Decomposition

Denote the parameter-efficient module $\mathbf{W} = \mathbf{B} \times \mathbf{A}$, $\mathbf{W} \in \mathbb{R}^{n \times n}$, and $\mathbf{W}$ as a low-rank matrix, *i.e.*, Rank($\mathbf{W}$) = $r$, $r \ll n$. The singular values of the matrix $\mathbf{W}$ are $\sigma_1 \geq \sigma_2 \geq \cdots \geq \sigma_r > 0$.

Based on singular value decomposition, the matrix $\mathbf{W}$ can be decomposed as:

$$\mathbf{W} = \boldsymbol{U\Sigma V}^T, \tag{6}$$

where $\boldsymbol{U} = [u_1, u_2, \cdots, u_r] \in \mathbb{R}^{n \times r}$ and $\boldsymbol{V} = [v_1, v_2, \cdots, v_r] \in \mathbb{R}^{n \times r}$ are orthogonal matrices with left and right normalized singular vectors, respectively, and $\boldsymbol{\Sigma} \in \mathbb{R}^{r \times r}$ is a diagonal matrix containing singular values of the original matrix, which can be formulated as:

$$\boldsymbol{\Sigma} = \begin{bmatrix} \sigma_1 & & & \\ & \sigma_2 & & \\ & & \ddots & \\ & & & \sigma_r \end{bmatrix}_{r \times r} \tag{7}$$

The low-rank matrix can therefore be rewritten as:

$$\begin{aligned} \mathbf{W} &= u_1 \sigma_1 v_1^T + u_2 \sigma_2 v_2^T + \cdots + u_r \sigma_r v_r^T \\ &= \sigma_1 u_1 v_1^T + \sigma_2 u_2 v_2^T + \cdots + \sigma_r u_r v_r^T. \end{aligned} \tag{8}$$

### A.2   Theoretical Analysis in Merging

For better illustration, we consider merging in a simplified situation and take two parameter-efficient modules as an example.

Let $\mathbf{W_1}, \mathbf{W_2}$ be two decomposed modules that are fine-tuned on task A and B, respectively:

$$\begin{aligned} \mathbf{W_1} &= \sigma_{11} u_{11} v_{11}^T + \sigma_{21} u_{21} v_{21}^T + \cdots + \sigma_{r1} u_{r1} v_{r1}^T, \\ \mathbf{W_2} &= \sigma_{12} u_{12} v_{12}^T + \sigma_{22} u_{22} v_{22}^T + \cdots + \sigma_{r2} u_{r2} v_{r2}^T, \end{aligned} \tag{9}$$

where $\sigma_{ij}, u_{ij}$ and $v_{ij}$ represent the $i^{th}$ singular value/left singular vector/right singular vector of the $j^{th}$ low-rank matrix, respectively.

Given the practical significance of singular vectors in LoRA, consider two permutations of 1 to $r$, *i.e.*, $(\bar{1}), (\bar{2}), \cdots, (\bar{r})$, and $(\underline{1}), (\underline{2}), \cdots, (\underline{r})$, merging the two modules can therefore be rewritten as:

$$\begin{aligned} \widetilde{\mathbf{W}} &= \lambda(\mathbf{W_1} + \mathbf{W_2}) \\ &= \lambda(\sigma_{(\bar{1})1} u_{(\bar{1})1} v_{(\bar{1})1}^T + \sigma_{(\bar{2})1} u_{(\bar{2})1} v_{(\bar{2})1}^T + \cdots + \sigma_{(\bar{r})1} u_{(\bar{r})1} v_{(\bar{r})1}^T \\ &\quad + \sigma_{(\underline{1})2} u_{(\underline{1})2} v_{(\underline{1})2}^T + \sigma_{(\underline{2})2} u_{(\underline{2})2} v_{(\underline{2})2}^T + \cdots + \sigma_{(\underline{r})2} u_{(\underline{r})2} v_{(\underline{r})2}^T) \\ &= \lambda\{(\sigma_{(\bar{1})1} u_{(\bar{1})1} v_{(\bar{1})1}^T + \sigma_{(\underline{1})2} u_{(\underline{1})2} v_{(\underline{1})2}^T) + (\sigma_{(\bar{2})1} u_{(\bar{2})1} v_{(\bar{2})1}^T + \sigma_{(\underline{2})2} u_{(\underline{2})2} v_{(\underline{2})2}^T) + \cdots \\ &\quad + (\sigma_{(\bar{r})1} u_{(\bar{r})1} v_{(\bar{r})1}^T + \sigma_{(\underline{r})2} u_{(\underline{r})2} v_{(\underline{r})2}^T)\}, \end{aligned} \tag{10}$$

where each pairwise subscript $\{(\bar{i}), (\underline{i})\}, i = 1, \cdots, r$ stands for similar singular components, *i.e.*, similar knowledge of two different matrices in low-rank space.

Empirically, $\boldsymbol{U}$ contains more general knowledge in low-rank space with larger similarity across tasks. Consequently, the merging process can mathematically be expressed as merging each of the task-specific vectors in low-rank space:

$$\lambda \sigma_{(\bar{i})1} v_{(\bar{i})1}^T + \lambda \sigma_{(\underline{i})2} v_{(\underline{i})2}^T = \lambda_{i1} \xi_{i1} + \lambda_{i2} \xi_{i2}, \quad i = 1, \cdots, r. \tag{11}$$

Given that $U, V$ are normalized, it can be inferred from Eqn. 11 that merging in low-rank space can be seen as vector addition for each group of task-specific singular vector, with singular vector $\xi_i$ indicating the direction and singular value $\sigma_i$ indicating the magnitude. Based on vector synthesis, direction change for task A and task B would be complementary to each other for each singular value.

It can be seen from the above derivation that due to the difference in original direction and magnitude, the singular values with larger magnitude are more likely to determine the direction and magnitude of the merged singular vector. As a result, the change in singular vector angle will vary from the perspective of singular value vectors belonging to different tasks due to the stark difference in singular values, *e.g.*, for task A, the direction angle change for vector 1 is small while the angle change for vector 2 is relatively large, and the situation is just the opposite for task B. Therefore, the key for merging would be maintaining direction robustness for vectors with small singular values. Without the loss of generality, the derivation can be extended to merging any number of models.

## B    Distribution of Singular Value in Different Layers

We depict the distribution of `attn.v` in Layer 1, 18, and 32 to show the distribution of singular value changes with different layers. It clearly shows in Fig. 8 that: (1) As the position of the layer becomes higher, the maximum singular value becomes larger, and the tail singular values become smaller, making the distribution more stark. This shares a similar observation with HiDe-LLaVA [8] that lower layers carry more general knowledge and higher layers contain more task-specific knowledge, so in the first layers, the gap between top and tail values would not be as large as in the last layers. Therefore, the distribution becomes more stark with increased layer, highlighting the necessity to properly handle direction instability during merging; (2) Moreover, compared with the original model, our method successfully and consistently scales both top and tail singular values across different layers, thereby contributing to robust and efficient merging with improved performance, which strongly demonstrates the effectiveness and rationality of the method.

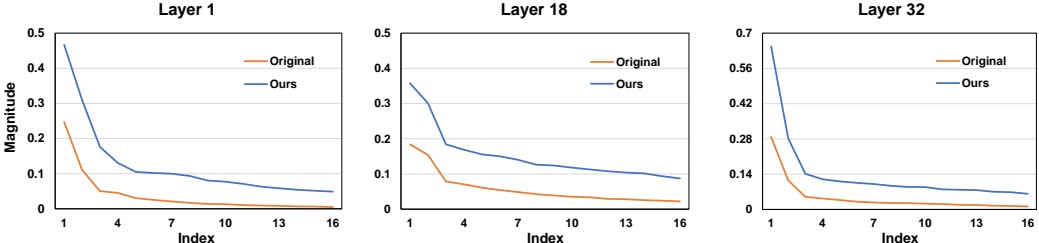

Figure 8: Singular value distribution of original and our model in `attn.v` of layer 1, 18 and 32.

## C    More Implementation Details

We build a multimodal codebase for multimodal tasks and vision tasks upon LLaVA [3] and CLIP [4], respectively. For training, we follow the standard training procedure described in LLaVA, *i.e.*, training each task individually and obtaining parameter-efficient modules. LoRA is added to linear layers in foundational blocks, and all models are trained for 1 epoch for merging.

For inference, the tasks are evaluated by accuracy. The vision task is conducted by constructing textual prompts for each category and calculating the similarity. Each evaluated sample is classified into the category with the largest similarity. Hyper-parameter $\lambda$ is set to 2 by default. All merging experiments are carried out on a single NVIDIA A6000 with the temperature set to 0.

## D    Details of Comparison Methods

The crucial operation of merging method involves designing a merging algorithm, *i.e.*, $\Phi(\cdot)$ defined in the paper. We primarily compare with DARE, Ties-merging, and Task Arithmetic in the main

---

[3]https://github.com/haotian-liu/LLaVA
[4]https://github.com/openai/CLIP

results. For comparison to these traditional approaches, we employ traditional approaches to LoRA components. Specifically, we fine-tune the foundation model using LoRA, merge LoRA components using traditional approaches, and finally evaluate performance with merged LoRA attached to base models. Their ways of merging are briefly introduced in the following paragraph.

**Task Arithmetic** views all parameters as vectors. It obtains task vectors by subtracting pre-trained models from fine-tuned models and performs standard arithmetic operations like addition and subtraction on them. It sums the parameters from checkpoints of different tasks and constructs a strong baseline for multi-task learning.

**Ties-merging** reduces the conflicts between parameters of different tasks by a trim, elect sign, and merging paradigm. Concretely, it first keeps parameters with the highest magnitudes, and then determines the aggregated sign based on the summation of remaining parameters. Finally, it merges the parameters with the same sign as the aggregated sign to mitigate disagreements.

**DARE** empirically observes the sparsity in parameters. It randomly drops parameters with a fixed ratio $p$, and rescales the remaining parameters with $1/(1-p)$ to match the expectation of parameters lost from dropping ones.

# E    Details of Different Tasks

## E.1    Composition of Instruction Tuning Datasets

The instruction tuning datasets follow the format of instruction tuning and are composed of image-text pairs and additional instruction templates. Instruction templates provide a clear and expressed task environment and purpose in natural language and are crucial for instruction tuning. The templates are shown in Appendix 8. In multimodal generative tasks, we carefully design the instruction template for each dataset. The templates are concatenated with task-specific inputs of image and text to the model to generate responses in an auto-regressive way. The language model is set to be trainable with the visual encoder frozen.

## E.2    Datasets of Vision Tasks

All the vision tasks are traditional datasets containing common objects across wide domains like cars, texture, traffic signs, and so on for image classification. Categories for them vary from 10 to 397. We fine-tune VLMs with LoRA on each task and merge them employing different types of merging methods. Only the visual encoder is trainable, and the text encoder remains frozen for label embedding extraction.

Table 8: Instruction templates for each dataset.

| Dataset | Instruction |
|---------|-------------|
| ScienceQA | Answer with the option's letter from the given choices directly. |
| ImageNet | What is the object in the image? Answer the question using a single word or phrase. |
| VQAv2 | Answer the question using a single word or phrase. |
| Grounding | Please provide the bounding box coordinate of the region this sentence describes: <description>. |
| IconQA | Answer the question using a single word or phrase. |
| VizWiz | What is happening in the image? Generate a brief caption for the image. |
| Flickr30k | What is happening in the image? Generate a brief caption for the image. |
| OCR-VQA | Answer the question using a single word or phrase. |
| AOKVQA | Answer with the option's letter from the given choices directly. |
| ImageNet-R | What is the object in the image? |
| Screen2Word | You are given a phone UI screen. Describe the screen in one sentence. |
| TabMWP | Answer the question using a single word or phrase. |

# F    More Experimental Results

**RobustMerge generalizes to the number of tasks.** We gradually increase the number of tasks to substantiate the robustness of our method. As is illustrated in Fig. 9, in seen tasks, the performance undergoes a slight drop as merging more models interferes with specific tasks. In unseen tasks, the performance first improves and then declines modestly. It could be attributed to the fact that in the first stage, seen tasks transfer knowledge and enhance unseen tasks of similar distribution; in the second stage, interference dominates merging rather than knowledge transformation. Under both circumstances, our method consistently outperforms existing approaches by a substantial margin as the task number varies, indicating the superiority and stability of our method.

Table 9: Results of merging models fine-tuned with DORA.

| Method | SciQA | Image | VQA | REC | OCR | VizWiz | Flickr | IconQA | **Average** |
|---|---|---|---|---|---|---|---|---|---|
| Task Arithmetic | 69.91 | 67.45 | 66.18 | 41.43 | 58.57 | 46.60 | 52.68 | 40.57 | 55.42 |
| Ties-merging | 69.01 | 64.06 | **66.60** | 40.68 | **61.94** | 46.51 | 51.97 | 35.82 | 54.57 |
| **RobustMerge** | **70.95** | **68.25** | 66.48 | **41.67** | 58.39 | **46.72** | **52.78** | **43.40** | **56.08** |

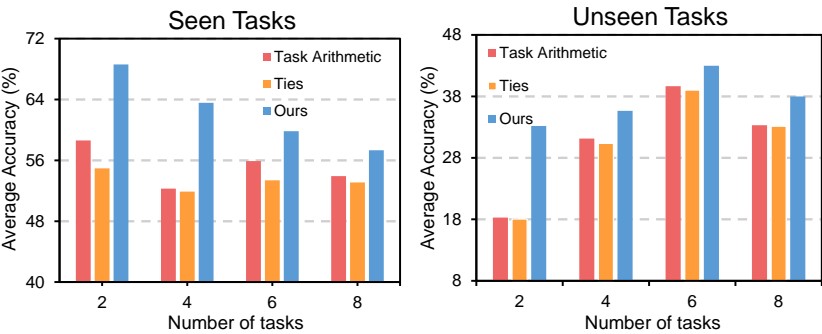

Figure 9: Average performance of seen and unseen tasks when number of tasks increases. Our method consistently outperforms TA and Ties with significant improvement.

**RobustMerge extends well to different PEFT methods.** We primarily test our method on LoRA since it is the most commonly used and comparable PEFT technique. To demonstrate the extensibility of the proposed method, we additionally conduct experiments on DoRA [34], which is a LoRA-based efficient technique with an advanced algorithm to improve the performance of PEFT learning. Results shown in Tab. 9 reveal that our method achieves consistent and substantial improvements against existing merging methods in different PEFT methods. It strongly certifies that the problem of direction robustness is widespread in merging different kinds of PEFT modules, where attention is primarily paid to improving the performance of a single task. By contrast, our method handles the issue to some extent, thereby promoting the multi-task performance when the PEFT technique varies.

