# OpenReview forum: "RobustMerge: Parameter-Efficient Model Merging for MLLMs with Direction Robustness"
_NeurIPS.cc/2025/Conference — NeurIPS 2025 spotlight_

### Official Review · Reviewer_e3Kd · 2025-06-19

**Clarity:** 3
**Significance:** 3
**Originality:** 3
**Rating:** 4
**Confidence:** 3

**Summary:**

The paper analyzes the directional robustness in the merging process from the perspective of low-rank decomposition, showing that compensating for the significant disparity between singular values helps improve directional robustness. It proposes a parameter-efficient merging method that does not require training, maintaining directional stability through complementary parameter adaptation. The paper prunes parameters based on their inter-relationships and scales the singular value coefficients to maintain directional stability and avoid interference between tasks. Experimental results demonstrate the superior performance and generalization ability of this method.

**Questions:**

1. What is the specific transformation applied to matrix B using the method of complementing parameter scaling in Section 3.3?
2. How does the method proposed in this paper compare with current model fusion methods such as Fisher[1], RegMean[2], BayesOpt[3], and LORE[4]?
[1] Merging Models with Fisher-Weighted Averaging
[2] Dataless Knowledge Fusion by Merging Weights of Language Models
[3] Checkpoint Merging via Bayesian Optimization in LLM Pretraining
[4] LORE-MERGING: Exploring Low-Rank Estimation For Large Language Model Merging

**Ethical Concerns:**

["NO or VERY MINOR ethics concerns only"]

**Final Justification:**

The author addressed my concerns about the applicability of other PEFT methods and validated the effectiveness of their approach. I will maintain my score.

**Limitations:**

yes

**Paper Formatting Concerns:**

No formatting issues

**Quality:**

3

**Strengths And Weaknesses:**

Strengths:
1. The paper proposes a novel Lora parameter merging method, which prunes parameters based on their inter-relationships and scales the singular value coefficients.
2. The paper conducts extensive experiments and analysis on different models and benchmarks, validating the effectiveness and generalization of the proposed method. It also performs a thorough set of ablation experiments.

Weaknesses:

1.Some of the baseline methods used for comparison are somewhat outdated. Comparing with more recent merging methods would help better highlight the effectiveness of the proposed method.
2. Can the merging method proposed in the paper be extended to other PEFT methods, such as PISSA, etc.?
3. No OOD results.

---

> ### Author Rebuttal · Authors · 2025-07-31
>
> Dear Reviewer e3Kd:
>
> Thank you for your effort in reviewing our paper and we are glad you are impressed by our analysis and thorough evaluation. We will respond to every question respectively.
>
> ### **Q1. Comparison with more fusion methods**
> Thank you for mentioning the valuable papers. Fisher[1] and RegMean[2] are popularly employed comparing methods for evaluation of model merging. They approximate the merging parameter by fisher matrix and finding closed-form solution for a least-squares regression problem, respectively. Both Fisher and RegMean need validation set to compute the coefficient of merged models, as shown in Tab.1, and our method refrains from validation set. BayesOpt[3] explores a new paradigm that merges checkpoints during pretraining to enhance downstream performance, while our method mainly targets at merging fine-tuned PEFT modules of different tasks. Since all models are fine-tuned with 1 epoch, [3] can not be directly applied to the setting. LORE-merging[4] iteratively solves a single-variable minimization problem with coordinate descent method to obtain better task vectors. By contrast, our method does not need the time-consuming iteration, with one straightforward calculation, which is much more efficient. As [4] does not provide its code, we can not reproduce satisfying results. Comprehensively considering reproducibility and comparability, we compare with Fisher[1] and RegMean[2] to better illustrate the effectiveness of the proposed method. Results are shown below.
>
> |Method|SciQA|ImgNet|VQAv2|Grounding|OCR|Viz|Flickr30k|IconQA|Avg
> |-|-|-|-|-|-|-|-|-|-
> |Fisher|69.17|56.21|64.05|38.61|58.74|42.35|46.79|38.06|51.74
> |RegMean|68.60|47.24|65.37|37.66|59.36|41.65|39.10|34.98|49.24
> |Ours|**73.43**|**65.54**|**67.20**|**44.80**|**62.97**|**46.61**|**52.80**|**45.90**|**57.33(+5.59)**
>
> It can be seen in the table that without validation set, our method still gets substantial promotion compared with Fisher and RegMean, strongly certificating the effectiveness of the proposed method. This could be attributed to in-depth analysis from the perspective of low-rank decomposition and specific merging algorithm design for PEFT components.
>
> We totally agree with the reviewer that comparison with recent merging methods would help highlight the effectiveness of the proposed method. Following the suggestions from the reviewer, we find that checkpoint merging[5, 6] would be a promising direction for merging techniques, and we would treat it as potential future direction. We promise to properly cite and add discussion with all these six valuable works in the updated manuscript.
>
> [1] Merging Models with Fisher-Weighted Averaging. NeurIPS 2022.
>
> [2] Dataless Knowledge Fusion by Merging Weights of Language Models. ICLR 2023.
>
> [3] Checkpoint Merging via Bayesian Optimization in LLM Pretraining. ICML 2025.
>
> [4] LORE-MERGING: Exploring Low-Rank Estimation For Large Language Model Merging.
>
> [5] Pruning via Merging: Compressing LLMs via Manifold Alignment Based Layer Merging. EMNLP 2024.
>
> [6] Towards enhanced LLM pretraining: Dynamic checkpoint merging via generation quality. Information Fusion.
>
> ### **Q2. Extension to other PEFT methods**
> Thank you for your constructive suggestion, comparison with more PEFT methods is significant to validate the extensibility of our method. In addition to PISSA[7], we also implement a commonly used PEFT method DoRA[8] that supports multimodal tasks. Due to time and character constraints, we conduct the experiment on a subset of 6 datasets and compare with two merging methods. Results are shown below.
>
> |Strategy|SciQA|ImgNet|VQAv2|Grounding|OCR|IconQA|Avg
> |-|-|-|-|-|-|-|-
> |LoRA+Task Arithmetic|71.92|59.78|66.31|40.51|63.54|42.10|57.36
> |+Ours|74.35|69.22|66.79|46.70|64.17|49.74|**61.82(+4.46)**
> |DoRA+Task Arithmetic|72.21|61.27|67.10|38.29|64.02|44.31|57.86
> |+Ours|74.89|69.40|67.14|44.24|64.85|52.26|**62.29(+4.26)**
> |PISSA+Task Arithmetic|72.36|60.89|67.43|39.11|64.51|45.99|58.38
> |+Ours|75.01|69.56|67.60|44.79|65.09|52.42|**62.57(+4.03)**
>
> It can be concluded from the results that **(1)** more advanced PEFT technique is beneficial to better performance of model merging, and exhibits overall better performance; **(2)** our method achieves consistent and substantial improvements against existing merging methods across all PEFT methods. It strongly certificates that **the problem of direction robustness widespread exists in merging all kinds of PEFT modules**, where attention is primarily paid to improving the performance of single task. By contrast, our method handles the issue to some extent, thereby promoting the performance when the PEFT technique varies.
>
> [7] PiSSA: Principal Singular Values and Singular Vectors Adaptation of Large Language Models. NeurIPS 2024.
>
> [8] DoRA: Weight-Decomposed Low-Rank Adaptation. ICML 2024.
>
> ### **Q3. OOD results**
> We firmly agree with the reviewer that validating OOD ability should be highlighted in model merging to certificate the generalization ability and extensibility of merging algorithm. In Tab.1, we use seen and unseen tasks to validate the generalization ability of merging model, where we merge eight fine-tuned models, and test them on four unseen tasks (line263-274). As can be seen from the table, **(1)** this can be viewed as OOD generalization that **existing methods perform poorly or even worse than zero-shot**, demonstrating its difficulty and that merging towards specific datasets would harm the generalization ability; **(2)** our method outperforms existing methods by a large margin, which could be attributed to the design of the merging method. Therefore, the generalizability of our proposed method is firmly demonstrated.
>
> Also, following the OOD setting of BayesOpt[3], we set VQAv2 as ID dataset and test merged models on three diverse OOD datasets: ScienceQA, IconQA and Grounding to further demonstrate the OOD ability. Results are shown below.
>
> |OOD Dataset|ScienceQA|IconQA|Grounding
> |-|-|-|-
> ID: VQAv2
> |Task Arithmetic|61.17|24.70|35.23
> |Ties|62.44|23.68|36.10
> |Ours|**64.09**|**27.91**|**39.46**
>
> Similar conclusion can be obtained from the additional experiment that our method achieves better OOD performance with less ID dataset, strongly demonstrating the effectiveness and generalization ability of the proposed method.
>
> ### **Q4: Form of specific transformation**
> The transformation matrix $W$ is applied between $A$ and $B$ in the form of matrix multiplication. As a result, the matrix can be seen operating on $A$ and $B$ simultaneously and no matrix operates independently on B. With respect to the specific form of the transformation matrix, it is a diagonal matrix based on statistical characteristics of $A$, which is defined in Eqn.(3) with detailed analysis in line203-214. Briefly speaking, considering that explicitly operating on decomposed matrices is time-consuming, we directly adjust original low-rank matrices to achieve the same goal inspired by the asymmetry and correlation between two LoRA modules. By designing this, singular values can be enlarged as illustrated in Fig.3a and Fig.3b.
>
> We once again thank the reviewer for time and effort in reviewing our paper with several insightful suggestions, and we hope the additional experiments and analyses could tackle your issues. We will properly cite the mentioned papers and add discussion in the updated manuscript. If you have any further questions, respond to us and we will try our best to address all your concerns!

---

> > ### Comment · Reviewer_e3Kd · 2025-08-07
> >
> > Thank you for the authors' detailed explanations and comprehensive experimental results

---

> > > ### Author Response · Authors · 2025-08-07
> > > **Thank You for Your Response**
> > >
> > > Dear reviewer e3Kd:
> > >
> > > Thank you for your response! We are really pleased to hear that our additional explanation and experiments help you have a better understanding of our paper! We will revise the manuscript according to your valuable feedback to further strengthen our paper. If you have any further questions, feel free to respond to us at your convenience! Thank you again for your time, effort and constructive participation in the discussion!
> > >
> > > Best wishes,
> > >
> > > Submission 2239 Authors.

---

### Official Review · Reviewer_5p9X · 2025-06-30

**Clarity:** 3
**Significance:** 3
**Originality:** 3
**Rating:** 5
**Confidence:** 4

**Summary:**

This paper introduces the method of CoPA-Merging, a training-free method for the merging of PEFT modules in multimodal large language models. The innovation of the method lies in addressing direction robustness. This happens due to large differences of singular values between LoRA modules. This is addressed through pruning and complementary parameter scaling to maintain proper stability of direction. It also does cross-task normalization to improve generalization to unseen tasks.

**Questions:**

How would the distribution of singular values change with different layers ?

How would the performance change if the number of merging models/loras change ?

How would the performance change if the merging was done differently for different layers ?

**Ethical Concerns:**

["NO or VERY MINOR ethics concerns only"]

**Final Justification:**

I think the authors have addressed my comments and I will increase the score.

**Limitations:**

Yes

**Quality:**

3

**Strengths And Weaknesses:**

---- Strengths ----

The concept of directional robustness is very new from the viewpoint of low rank adaptation merging.

The great thing is that is training-free and is efficient and it does not require additional training, storage etc.

It shows strong empirical results and outperforms baselines like Task Arithmetic, DARE and Ties-Merging.

There has been thorough evaluation on seen and unseen tasks.

Analysis and Mathematical rigor is well defined in the form of variation in singular values.

---- Weakness ----

The method has only been carried on LoRA and not other PEFT tasks.

The method has only been tried on multimodal tasks. It has not been applied to image generation tasks.

---

> ### Author Rebuttal · Authors · 2025-07-31
>
> Dear Reviewer 5p9X:
>
> Thank you for your professional suggestions and we greatly appreciate your positive comments about our analysis and empirical results. We will respond to each question respectively.
>
> ### **Q1. More PEFT methods**
> Thank you for your valuable advice. Evaluating the method on different PEFT modules is of great significance to demonstrate the extensibility of the proposed method. We primarily test our method on LoRA since it is the most commonly used and comparable PEFT techniques. Following your suggestions, we additionally conduct experiments DoRA[1] and PISSA[2], which are two LoRA-based efficient techniques with advanced algorithms to improve the performance of PEFT learning, to demonstrate the extensibility of the proposed method. Detailed results and analyses are shown below.
>
> |Strategy|SciQA|ImgNet|VQAv2|Grounding|OCR|IconQA|Avg
> |-|-|-|-|-|-|-|-
> |LoRA+Task Arithmetic|71.92|59.78|66.31|40.51|63.54|42.10|57.36
> |+Ours|74.35|69.22|66.79|46.70|64.17|49.74|**61.82(+4.46)**
> |DoRA+Task Arithmetic|72.21|61.27|67.10|38.29|64.02|44.31|57.86
> |+Ours|74.89|69.40|67.14|44.24|64.85|52.26|**62.29(+4.26)**
> |PISSA+Task Arithmetic|72.36|60.89|67.43|39.11|64.51|45.99|58.38
> |+Ours|75.01|69.56|67.60|44.79|65.09|52.42|**62.57(+4.03)**
>
> It can be concluded from the results that **(1)** more advanced PEFT technique is beneficial to better performance of model merging, and exhibits overall better performance; **(2)** our method achieves consistent and substantial improvements against existing merging methods across all PEFT methods. It strongly certificates that **the problem of direction robustness widespread exists in merging all kinds of PEFT modules**, where attention is primarily paid to improving the performance of single task. By contrast, our method handles the issue to some extent, thereby promoting the performance when the PEFT technique varies.
>
> [1] DoRA: Weight-Decomposed Low-Rank Adaptation. ICML 2024.
>
> [2] PiSSA: Principal Singular Values and Singular Vectors Adaptation of Large Language Models. NeurIPS 2024.
>
> ### **Q2. More task types**
> We truly appreciate the reviewer for the professional suggestion. Due to the large gap between image generation and understanding task, we do not find existing merging methods that conduct additional experiments on image generation task. To this end, we conduct experiments on vision tasks with VLM model to partially demonstrate the extensibility of our approach. With respect to image generation task, the most related work would be ZipLoRA[3], which focuses on style transfer with merging strategies. It trains a content LoRA and a style LoRA and then merges them to get one LoRA with both style and content knowledge. However, it **requires post-training merged LoRAs with specific design**.
>
> As the reviewer suggests, we conduct experiments on style transfer to validate the extensibility of the proposed method in image generation task. Following the setting from ZipLoRA[3], we train a content LoRA and a style LoRA based on SDXL. We calculate the feature similarity of subject, style and text between specific and merged models to reflect the alignment and fidelity of the merged model. Results are shown below.
>
> |Method|Style-alignment $\uparrow$|Subject-alignment $\uparrow$|Text-alignment $\uparrow$
> |-|-|-|-
> |Direct Merge|0.472|0.271|0.213
> |Task Arithmetic|0.474|0.269|0.233
> |Ties|0.481|0.274|0.231
> |Ours|0.484|0.288|0.240
>
> Due to the time limitation and merging without post-training, the model might not be that converged with overall smaller similarity in value, but the advantage of our method can still be confirmed that our method outperforms existing merging techniques in evaluation of all style, subject and text alignment, thereby showing the extensibility of the proposed merging method.
>
> [3] ZipLoRA: Any Subject in Any Style by Effectively Merging LoRAs. ECCV 2024.
>
> ### **Q3. Singular value distribution of different layers**
>
> We take merging ScienceQA and ImgetNet in Fig.3 as an example to show the distribution of singular value changes with different layers. Due to the constraint of the policy, we are only able to present the results in table, so we present average singular value first layer, middle layer and last layer to represent the overall change in distribution of singular values. We select the maximum, medium and minimum of singular value to show the distribution. Results are shown in table below.
>
> |Layers|First|Middle|Last|
> |-|-|-|-
> Original
> |Maximum singular value|0.1177|0.1645|0.2772
> |Minimum singular value|0.0183|0.0173|0.0142
> |Medium singular value|0.0328|0.0272|0.0179
> Ours
> |Maximum singular value|0.2102|0.2841|0.4842
> |Minimum singular value|0.0284|0.0262|0.0205
> |Medium singular value|0.0426|0.0311|0.0244
>
> From the table above, it can be concluded that:
>
> **(1)** as the **position of the layer becomes higher, the maximum singular value becomes larger, and the tail singular values become smaller** (reflected by both smaller minimum and medium singular values), making the distribution more stark. This shares similar observation with HiDe-LLaVA[4] that lower layers carry more general knowledge and higher layers contain more task-specific knowledge, so in the first layers, gap between top and tail values would not as large as the last layers (This could also be validated by the results of merging different layers in **Q5** that naive merging of higher layers leads to more performance drop). Therefore, the distribution becomes more stark with increased layer, highlighting the necessity to properly handle direction instability during merging.
>
> **(2)** Moreover, comparing our model with original model, we can conclude that **our method successfully and consistently scales both top and tail singular values across different layers**, thereby contributing to robust efficient merging with improved performance, which strongly demonstrates the effectiveness and rationality of the proposed method.
>
> It would be clearer to present the results in the form of figure, and we would properly display it in the updated manuscript.
>
> ### **Q4. Different number of merging models**
> We provide results of merging different number of models in Fig.A of Appendix. It can be concluded that **(1)** in seen tasks, the performance undergoes slight drop, which is in consistent with the intuition that merging more models interferes specific task more; **(2)** in unseen tasks, the performance first improves and then declines modestly. The phenomenon could be attributed that in the first stage, seen tasks transfer knowledge and enhance unseen tasks of similar distribution; in the second stage, interference dominates merging rather than knowledge transformation with increasing merging models; **(3)** It is notable that under both circumstances, our method consistently and substantially outperforms existing approaches as task number varies, indicating the superiority and stability of our method.
>
> ### **Q5. Merging different layers**
> To get the influence of merging different layers, we conduct experiments on merging low, middle and high layers of the model to observe the performance variation. Specifically, we perform the proposed method on selected layers and naive average merging on the rest layers. The results are shown in table below.
>
> |Merging layers of our approach|SciQA|ImgNet|VQAv2|Grounding|OCR|Viz|Flickr30k|IconQA|Avg
> |-|-|-|-|-|-|-|-|-|-
> |First 8 layers|70.85|58.69|67.66|39.40|**65.25**|43.48|46.41|31.48|52.90(-4.43)
> |Middle 8 layers|71.44|58.32|66.65|39.22|64.60|44.44|48.50|37.29|53.80(-3.53)
> |Last 8 layers|71.64|56.50|66.86|42.07|63.76|46.16|51.40|35.71|54.26(-3.07)
> |Full (Ours)|**73.43**|**65.54**|**67.20**|**44.80**|62.97|**46.61**|**52.80**|**45.90**|**57.33**
>
> It can be inferred from the results that **(1)** As observed from **Q3** that higher layers exhibit more stark distribution, **naive merging higher layers is more prone to poorer performance** (reflected by the first two rows). This shares similar conclusion with results from **Q3** and HiDe-LLaVA[4] that lower layers carry more general knowledge and higher layers contain more task-specific knowledge. As the specific knowledge dominates the distribution, the rest singular values are more likely to alter their directions during merging. Therefore, merging higher layers without specific design inevitably leads to direction instability. Therefore, when operating our approach on the higher layers (Last 8 layers in table), performance would be the best compared to naive merging of highest layers. This is of great significance that it gives a clue for which layers are more prone to direction instability during merging; **(2)** Our approach conducts from the perspective of direction robustness. and all layers merging with our method achieves the best results, furthermore certificating the effectiveness of the proposed merging method.
>
> [4] HiDe-LLaVA: Hierarchical Decoupling for Continual Instruction Tuning of Multimodal Large Language Model. ACL 2025.
>
> We would like to express our sincere gratitude to the reviewer for all the thoughtful suggestions, and we hope these additional experiments could address the concerns and better strengthen the proposed method. We will update the manuscript with additional citations, experiments and analyses. If you have any remaining questions, do not hesitate to respond to us, and we are pleased to address any of your concerns!

---

### Official Review · Reviewer_Rk6Q · 2025-07-01

**Clarity:** 3
**Significance:** 3
**Originality:** 3
**Rating:** 5
**Confidence:** 4

**Summary:**

This paper focuses on the challenge of merging multiple task-specific parameter-efficient models into a single high-performance and generalizable multimodal large language model. The authors introduce CoPA-Merging, a training-free merging framework that maintains direction robustness in low-rank space through adaptively scaling singular values. Comprehensive experiments demonstrate CoPA-Merging’s effectiveness in maintaining multi-task performance across seen and unseen domains.

**Questions:**

Please refer to Weaknesses.

**Ethical Concerns:**

["NO or VERY MINOR ethics concerns only"]

**Final Justification:**

After the author rebuttal, all of my concerns have been addressed. I think the paper provides a novel insight for keeping the robustness while merging model. The technique is reasonable and the experiments are comprehensive. I therefore suggest accept.

**Limitations:**

Limitations are discussed in Appendix.

**Paper Formatting Concerns:**

No Concerns.

**Quality:**

3

**Strengths And Weaknesses:**

**Strengths:**

1. Efficient merging is an important topic for current large-scale pre-trained models.

2. The analysis from the perspective of direction robustness of singular values is interesting and reasonable for multi-task merging.

3. Comprehensive experiments well demonstrate the effectiveness of CoPA-Merging.

**Weaknesses:**

I think the motivation is sufficient and the experiments well support the claims, so I only have several minor suggestions that may further strengthen the paper in my mind.

1.  More PEFT methods. The exploration mainly focuses on LoRA-based mechanisms. How does it work on other PEFT methods, such as P-Tuning [1] or LLaMA-Adapter [2]?

2. More MLLMs. Only LLaVA is employed in the paper. How does CoPA-Merging perform on other MLLMs, such as Qwen2-VL [3]?

3. More analysis or visualization on task relationships. The paper mentions that the small singular values of one task may be altered by other tasks, reducing the performance. Is there experimental evidence indicating this phenomenon, and are there direct visualizations or analyses showing how CoPA-Merging mitigates it?

***References:***

[1] Xiao Liu et al., “P-Tuning: Prompt Tuning Can Be Comparable to Fine-tuning Across Scales and Tasks”, ACL, 2022.

[2] LLaMA-Adapter-V2 Multi-modal, https://github.com/OpenGVLab/LLaMA-Adapter/tree/main/llama_adapter_v2_multimodal7b, 2024.

[3] Peng Wang et al., “Qwen2-VL: Enhancing Vision-Language Model's Perception of the World at Any Resolution”, arXiv, 2024.

---

> ### Author Rebuttal · Authors · 2025-07-31
>
> Dear Reviewer Rk6Q:
>
> Thank you for your professional comments about our paper and recognition about the analysis from direction robustness! We will answer each question respectively.
>
> ### **Q1. More PEFT methods**
> Our motivation is built upon matrix analysis of singular values and design specific techniques to resolve interference of direction robustness when merging LoRA components, which is based on the consideration that LoRA has been the most popular and effective parameter-efficient tuning technique across various fields. Therefore, it can not be directly applied to prompt-based methods. In this part, we do not know any existing work about merging prompts to the best of our knowledge. Nevertheless, we respectully agree with your suggestion that validating the extensibility across different PEFT methods is beneficial to further demonstrate the effectiveness of the method. Therefore, we give some additional results on other PEFT techniques like PISAA[1], DoRA[2], which are LoRA-based efficient modules with advanced techniques to improve the effectiveness of PEFT learning, to demonstrate the extensibility of the proposed method. Detailed results and analyses are shown below.
>
>
> |Strategy|SciQA|ImgNet|VQAv2|Grounding|OCR|IconQA|Avg
> |-|-|-|-|-|-|-|-
> |LoRA+Task Arithmetic|71.92|59.78|66.31|40.51|63.54|42.10|57.36
> |+Ours|74.35|69.22|66.79|46.70|64.17|49.74|**61.82(+4.46)**
> |DoRA+Task Arithmetic|72.21|61.27|67.10|38.29|64.02|44.31|57.86
> |+Ours|74.89|69.40|67.14|44.24|64.85|52.26|**62.29(+4.26)**
> |PISSA+Task Arithmetic|72.36|60.89|67.43|39.11|64.51|45.99|58.38
> |+Ours|75.01|69.56|67.60|44.79|65.09|52.42|**62.57(+4.03)**
>
> It can be concluded from the results that **(1)** more advanced PEFT technique is beneficial to better performance of model merging, and exhibits overall better performance; **(2)** our method achieves consistent and substantial improvements against existing merging methods across all PEFT methods. It strongly certificates that **the problem of direction robustness widespread exists in merging all kinds of PEFT modules**, where attention is primarily paid to improving the performance of single task. By contrast, our method handles the issue to some extent, thereby promoting the performance when the PEFT technique varies.
>
> We firmly agree with your opinion and will treat exploration to more types of parameter efficient tuning merging as potential future direction. We will cite and add discussion about this in the updated manuscript.
>
> [1] PiSSA: Principal Singular Values and Singular Vectors Adaptation of Large Language Models. NeurIPS 2024.
>
> [2] DoRA: Weight-Decomposed Low-Rank Adaptation. ICML 2024.
>
> ### **Q2. More experiments on other MLLMs**
> Thank you for your constructive suggestions. We firmly agree with you that validating on other MLLMs is crucial for the extensibility of the proposed method. Following your suggestion, we conduct additional experiments on **Qwen-VL** and the results are shown below.
>
> |Method|SciQA|ImgNet|VQAv2|Grounding|OCR|Viz|Flickr30k|IconQA|Avg
> |-|-|-|-|-|-|-|-|-|-
> |Task Arithmetic|66.02|54.20|67.74|29.76|64.77|47.10|49.01|38.57|52.14
> |Ties|66.27|54.06|67.02|30.31|**66.90**|46.81|46.65|33.15|51.39
> |Ours|**69.45**|**63.91**|**69.02**|**35.61**|65.04|**49.28**|**51.89**|**43.06**|**55.90(+3.76)**
>
> It is indicated that compared with existing merging strategies, our method also achieves significant improvement on Qwen-VL (**+3.76%**), strongly demonstrating its effectiveness across different architectures.
>
> ### **Q3. More analysis on task relationships**
> We totally agree with you that demonstrating the phenomenon of task interference would further strengthen the effectiveness of the method. To this end, would explain it from three aspects.
>
> **1)** We calculate the **average similarity of each corresponding singular vector** between task-specific models and merged models to reflect the direction deviation in merging. Larger similarity means the direction possesses more robustness and is not prone to change direction during merging, thereby maintaining performance of specific task. Moreover, we also incorporate **ratio of singular value** between merged and original models to comprehensively reflect the degree of specific knowledge learning.
>
> Empirically taking merging ScienceQA and ImageNet in Fig.3b as an example, we present similarity of singular vector and ratio of singular value between task-specific and merged model. As is analyzed in the paper (Fig.3a and Fig.3b), the largest singular value displays more direction robustness, so we divide the value into largest and average of remaining parts for better illustration. Results are shown below.
>
> |Method|Similarity of largest singular vector $\uparrow$ | Average similarity of remaining singular vectors $\uparrow$|Ratio of largest singular value $\uparrow$| Average ratio of remaining singular values $\uparrow$
> |-|-|-|-|-
> |Task Arith|0.63|0.16|0.6|1.3
> |Ours|**0.67**|**0.27**|**1.3**|**1.8**
>
> It can be concluded that **(1)** during merging, **the largest vector tends to be stable, while remaining vectors are extremely dissimilar** (direction instability), which leads to performance drop in evaluation. By contrast, our method substantially improves the similarity of remaining vectors (**x1.7**), strongly promoting merging performance (line144-146); **(2)** Moreover, ratio of value also reflects that compared to specific model, existing method would decrease the largest singular value and fail to sufficiently strengthen smaller singular values. By contrast, our method better **enhances smaller values and maintains specific knowledge during merging**, which is in consistent with the paper that scaling smaller values contributes to direction robustness (line153-157).
>
> **2)** From the definition established above, we additionally conduct ablation study about the similarity to certificate the effectiveness of the proposed pruning and scaling, and the results are shown below.
>
> |Method|Similarity of largest singular vector $\uparrow$ | Average similarity of remaining singular vectors $\uparrow$|Ratio of largest singular value $\uparrow$| Average ratio of remaining singular values $\uparrow$
> |-|-|-|-|-
> |Task Arithmetic|0.63|0.16|0.6|1.3
> |Prune|0.62|0.20|0.5|1.2
> |Prune&Scale (Ours)|**0.67**|**0.27**|**1.3**|**1.8**
>
> These analyses give a more clear explanation about the function of each component and how our method mitigates the problem of direction instability: (1) Prune is to **resolove the interference between tasks while exhibiting the least influence on the direction** (line189-192), and the sparsification also boosts the robustness of small values (two opposite conflicting parameters would be mitigated by setting one value to zero); (2) Equipping scale after prune aims to **compensate for singular value drop raised by pruning** (line 201-203), thereby enhancing the direction robustness, which can be reflected by both similarity and ratio increase.
>
> Quantitative experiments are as follows (Tab.6 in paper) to intuitively display performance gain from the two techniques.
>
> |Component|SciQA|ImgNet|VQAv2|Grounding|OCR|Viz|Flickr30k|IconQA|Avg
> |-|-|-|-|-|-|-|-|-|-
> |Task Arithmetic|71.94|57.49|67.06|38.90|62.87|44.80|49.20|39.21|53.93
> |Prune|73.03|64.18|**67.50**|43.12|58.19|46.36|52.24|44.54|56.14(+2.21)
> |Prune&Scale (Ours)|**73.43**|**65.54**|67.20|**44.80**|**62.97**|**46.61**|**52.80**|**45.90**|**57.33(+3.40)**
>
> **3)** We furthermore give an analysis from the perspective of distribution of singular values in different layers in **Q3 of Reviewer 5p9X** to demonstrate the effectiveness of our method. The summarized conclusion is that **higher layers exhibit more stark distributions**, which means larger gap between task-specific knowledge with more direction variation, highlighting the need for handling the issue of direction instability, and our method designed for direction robustness well mitigates the problem.
>
> Unfortunately, we are not able to present a picture to display it due to the constraint of the policy. We show qualitative results above and we will add extensive analyses detailedly into the updated manuscript to strengthen our method.
>
> We express our respectful gratitude to the reviewer for your time and effort, and recognition of our paper. We hope the various additional experiments could strengthen our paper and we will revise the paper according to your valuable feedback. If you have any remaining questions or think some experiments are insufficient, feel free to respond to us at your convenience and we will do our best to address all the concerns!

---

> > ### Comment · Reviewer_Rk6Q · 2025-08-02
> > **Response to Author Rebuttal**
> >
> > Thank the reviewers for the careful and detailed responses. After the rebuttal, all of my concerns have been well addressed, and I have raised my rating.

---

> > > ### Author Response · Authors · 2025-08-02
> > > **Great Appreciation for your Recognition**
> > >
> > > Dear Reviewer Rk6Q:
> > >
> > > Thank you for your response! We are really glad that our additional experiments and explanation help you have a better understanding of our paper and all of your concerns are successfully addressed!
> > >
> > > We greatly appreciate your recognition and we are sincerely grateful that you raise the rating! We will update the manuscript based on the discussion to strengthen our paper. Thank you again for your affirmation and active participation in the discussion!
> > >
> > > Best wishes,
> > >
> > > Submission 2239 Authors.

---

### Official Review · Reviewer_f2pm · 2025-07-10

**Clarity:** 3
**Significance:** 3
**Originality:** 3
**Rating:** 5
**Confidence:** 3

**Summary:**

In this paper authors look at the problem of merging parameter-efficient fine-tuned (PEFT) models. Typically, model merging approaches fail under PEFT settings. Authors claim the reason of poor performance as direction instability along directions of small singular values. Authors propose to resolve this by 1) pruning smallest magnitude weights and  2) renormalizing the PEFT A matrix.
Authors empirically show that this procedure leads to improvement in the model merging performance.

**Questions:**

- It’ll be helpful to resolve the weaknesses in the paper

**Ethical Concerns:**

["NO or VERY MINOR ethics concerns only"]

**Final Justification:**

I thank the authors for clarification in the rebuttal and the proposed metrics for measuring directional robustness. Given the novelty in addressing PEFT model merging, and the promising results across multiple benchmarks, I lean toward acceptance.

**Limitations:**

Yes.

**Paper Formatting Concerns:**

No concerns.

**Quality:**

3

**Strengths And Weaknesses:**

Strengths:
- The paper looks at an important problem merging PEFT models and proposes a novel methodology (CoPA merging) for solving the problem.
- Thorough empirical evaluation on VLMs with detailed ablation studies.
- Results in Table 4 and 5 are promising and CoPA merging leads to improvement in model merging

Weaknesses:
- Even though authors identify directional robustness as a critical property for achieving model merging for PEFT models, there is no formal definition and they lack evaluation of when one can identify if PEFT models are robust or not. It needs to be formalized
- I can’t seem to understand how pruning and scaling lead to directional robustness so justification is not entirely clear to me. I think this is related to not having a proper definition established in the paper.

---

> ### Author Rebuttal · Authors · 2025-07-31
>
> Dear Reviewer f2pm:
>
> We sincerely thank you for your time and effort in reviewing our paper with several insightful suggestions, and we will respond to each question respectively.
> ### **Q1 Evaluation of robustness of PEFT models**
> Thank you for your constructive suggestion. We totally agree that formulating the robustness of specific model contributes to better understanding of the effectiveness of merging method. To better illustrate this part, we use the **average similarity of each corresponding singular vector** between task-specific models and merged models as the criterion, which reflects the direction deviation in merging. Larger similarity means the direction possesses more robustness and is not prone to change direction during merging, thereby maintaining performance of specific task. Moreover, we also incorporate the **ratio of singular value** between merged and original models to comprehensively reflect the degree of specific knowledge learning.
>
> Empirically taking merging ScienceQA and ImageNet in Fig.3b as an example, we present similarity of singular vector and ratio of singular value between task-specific model and merged model and compare two merging methods. As is analyzed in the paper (Fig.3a and Fig.3b), the largest singular value displays more direction robustness, so we divide the value into largest and average of remaining parts for better illustration. Results are shown below.
>
> |Method|Similarity of largest singular vector $\uparrow$ | Average similarity of remaining singular vectors $\uparrow$|Ratio of largest singular value $\uparrow$| Average ratio of remaining singular values $\uparrow$
> |-|-|-|-|-
> |Task Arithmetic|0.63|0.16|0.6|1.3
> |Ours|**0.67**|**0.27**|**1.3**|**1.8**
>
> It can be concluded that **(1)** during merging, **the largest vector tends to be stable, while remaining vectors are extremely dissimilar** (direction instability), which leads to performance drop in evaluation. By contrast, our method substantially improves the similarity of remaining vectors compared to Task Arithmetic (**x1.7**), strongly promoting merging performance (line144-146); **(2)** Moreover, ratio of value also reflects that compared to specific model, existing method would decrease the largest singular value and fail to sufficiently strengthen smaller singular values. By contrast, our method **better enhances smaller values and maintains knowledge during merging**, which is in consistent with the paper that scaling smaller values contributes to direction robustness (line153-157).
>
> ### **Q2: Intuitive contribution of each component**
> From the definition established above, we conduct ablation study about the similarity to certificate the effectiveness of the proposed pruning and scaling, and the results are shown below.
>
> |Method|Similarity of largest singular vector $\uparrow$ | Average similarity of remaining singular vectors $\uparrow$|Ratio of largest singular value $\uparrow$| Average ratio of remaining singular values $\uparrow$
> |-|-|-|-|-
> |Task Arithmetic|0.63|0.16|0.6|1.3
> |Prune|0.62|0.20|0.5|1.2
> |Prune&Scale (Ours)|**0.67**|**0.27**|**1.3**|**1.8**
>
> These analyses give a more clear explanation about the function of each component: **(1)** Prune is to **resolove the interference between tasks while exhibiting the least influence on the direction** (line189-192), and the sparsification also boosts the robustness of small values (two opposite conflicting parameters would be mitigated by setting one value to zero); **(2)** Equipping scale after prune aims to **compensate for singular value drop raised by pruning** (line 201-203), thereby enhancing the direction robustness, which can be reflected by both similarity and ratio increase.
>
> Quantitative experiments are as follows (Tab.6 in paper) to intuitively display performance gain from the two techniques.
>
> |Component|SciQA|ImgNet|VQAv2|Grounding|OCR|Viz|Flickr30k|IconQA|Avg
> |-|-|-|-|-|-|-|-|-|-
> |Task Arithmetic|71.94|57.49|67.06|38.90|62.87|44.80|49.20|39.21|53.93
> |Prune|73.03|64.18|**67.50**|43.12|58.19|46.36|52.24|44.54|56.14(+2.21)
> |Prune&Scale (Ours)|**73.43**|**65.54**|67.20|**44.80**|**62.97**|**46.61**|**52.80**|**45.90**|**57.33(+3.40)**
>
>
> Due to the constraint of the policy, we are only able to present table instead of better-illustrated figures. We will add the additional analysis properly and detailedly into updated manuscripts to strengthen our method.
>
> We thank again for your insightful and professional comments. We hope the additional definition and analysis can help you have a better understanding of how our method contributes to direction robustness. If you have any remaining question that needs to be further clarified, do not hesitate to respond to us and we will try our best to address all the concerns!

---

### Note · Authors · 2025-08-11

Dear Area Chair:

We sincerely appreciate the effort of all reviewers and AC in reviewing our paper. We are really glad that our paper receives recognition and positive reviews from all reviewers. We thank all the reviewers for and highly recognizing our paper as "**reasonable analysis from the perspective of direction robustness**" (Rk6Q, 5p9X), "**novel merging method**" (f2pm, e3Kd) "**thorough evaluation with detailed ablation**" (f2pm, 5p9X, e3Kd), "**significant improvement in merging**" (f2pm, Rk6Q, 5p9X, e3Kd).

During the discussion, we conducted various experiments with detailed explanations following the valuable suggestions from the reviewers, and successfully addressed all concerns from two participated reviewers commented as "**careful and detailed responses**" (Rk6Q), "**detailed explanations and comprehensive experimental results**" (e3Kd). As all reviewers hold positive attitude toward our paper with few questions and no more questions are raised during the discussion periods, we believe the additional experiments and analyses in our rebuttal sufficiently address all concerns from reviewers, and we will carefully update the manuscript according to the discussion.

We respectfully hope you can make the decision taking these into consideration. Thank you again for your time and effort.

Best regards,

Submission 2239 Authors.

---

### Decision · Program_Chairs · 2025-09-17

**Decision:**

Accept (spotlight)

**Comment:**

The paper received (A, A, A, BA) with confidence (3, 4, 4, 3).  All reviewers agreed that the work tackles an important and timely problem, is novel, and shows strong empirical performance with extensive ablations. The novelty of the methodology and the empirical verification are convincing, and the consensus  after rebuttal was positive. The decision is to accept.